# Oleanolic Acid: Extraction, Characterization and Biological Activity

**DOI:** 10.3390/nu14030623

**Published:** 2022-01-31

**Authors:** José M. Castellano, Sara Ramos-Romero, Javier S. Perona

**Affiliations:** 1Department of Food and Health, Instituto de la Grasa-CSIC, Campus of the University Pablo de Olavide, Building 46, 41013 Seville, Spain; jmcas@ig.csic.es (J.M.C.); perona@ig.csic.es (J.S.P.); 2Department of Cell Biology, Physiology and Immunology, Faculty of Biology, University of Barcelona, Av. Diagonal 643, 08028 Barcelona, Spain

**Keywords:** oleanolic acid, structure, characterization, biological activities, mechanisms, diseases

## Abstract

Oleanolic acid, a pentacyclic triterpenoid ubiquitously present in the plant kingdom, is receiving outstanding attention from the scientific community due to its biological activity against multiple diseases. Oleanolic acid is endowed with a wide range of biological activities with therapeutic potential by means of complex and multifactorial mechanisms. There is evidence suggesting that oleanolic acid might be effective against dyslipidemia, diabetes and metabolic syndrome, through enhancing insulin response, preserving the functionality and survival of β-cells and protecting against diabetes complications. In addition, several other functions have been proposed, including antiviral, anti-HIV, antibacterial, antifungal, anticarcinogenic, anti-inflammatory, hepatoprotective, gastroprotective, hypolipidemic and anti-atherosclerotic activities, as well as interfering in several stages of the development of different types of cancer; however, due to its hydrophobic nature, oleanolic acid is almost insoluble in water, which has led to a number of approaches to enhance its biopharmaceutical properties. In this scenario, the present review aimed to summarize the current knowledge and the research progress made in the last years on the extraction and characterization of oleanolic acid and its biological activities and the underlying mechanisms of action.

## 1. Introduction

Oleanolic acid (OA) is a ubiquitous pentacyclic triterpenoid in the plant kingdom, often used as medicinal herbs, and is an integral part of the human diet. Only in the 2010–2021 period, the major international databases have recorded several thousand publications dealing with oleanolic acid (15,818 and 3261 entries for Scopus and PubMed, respectively), reflecting the great interest of the scientific community and the progress in the understanding of this biomolecule. These reports include the isolation and purification of the triterpene from a plethora of plants and herbs, the pharmacological research on its beneficial effects, bioavailability and toxicity studies and the clinical use of OA in the prevention of different pathologies, mainly those chronic disorders in which oxidative stress and inflammation are underlaying mechanisms. Furthermore, significant research has been developed on the chemical modification of its molecule to obtain more effective and soluble synthetic derivatives. This work shows an updated review of the main milestones achieved in the extraction and characterization of naturally occurring OA, and the latest contributions to the knowledge of its biological activity.

## 2. Oleanolic Acid Characterization

### 2.1. Names, Structure and Identifiers

Oleanolic acid (C30H48O3), (3b-hydroxy-olean-12-en-28-oic acid; CAS 508-02-1; PubChem CID 10494; EC number 200-081-6; NCS number 114945; IUPAC name 4aS,6aR,6aS,6bR,8aR,10S,12aR,14bS)-10-hydroxy-2,2,6a,6b,9,9,12a-heptamethyl-1,3,4,5,6,6a,7,8,8a,10,11,12,13,14b-tetradecahydropicene-4a-carboxylic acid) (Figure 1) is a pentacyclic triterpenoid, conjugate acid of an oleanolate, that occurs widely in many plants as the free acid or the aglycone for many saponins and plays a role as a secondary metabolite. OA is biosynthesized from lupine and can rearrange to its isomer, ursolic acid or be oxidized to taraxasterol and amyrin. OA gives positive the Liberman–Noller test, developing yellow color with tetranitromethane (TNM), which indicates the triterpenoid nature of the molecule. In addition, it does not respond to Molisch’s test, suggesting a non-glycosidic nature. It also forms mono acetate with acetic anhydride–pyridine suggesting the presence of one hydroxyl group in the molecule.

### 2.2. Chemical and Physical Properties

The molecular weight of OA has been determined as 456.7 g/mol. OA is a hard-hydrophobic compound, practically insoluble in water (1.748 µg/L), which is usually extracted from plants using organic solvents such as methanol, ethanol, 1-butanol, ethyl acetate, diethyl ether or acetone.

Pure oleanolic acid appears as a crystalline solid that exhibits polymorphism. When directly crystalizing from plant extracts (for instance, ethanolic extracts from olive leaf), OA appears in the form of fine, whitish prismatic crystals (Figure 2), whereas the OA recrystallized from methanol, ethanol or acetone appears as fine needle-shaped crystals [1,2]. On the other hand, when OA is crystallized by fast cooling or fast concentration from chloroform or dichloromethane, it exhibits an amorphous nature [1].

Albi et al. determined in 2001 [3] the melting point of OA using differential scanning calorimetry (DSC). To obtain the thermogram, they introduced the compound sample into a sealed aluminum capsule and executed a temperature program from 60 to 320 °C at 40 °C/min. Indium (Tm = 156.6 °C; ΔHf = 28.45 J/g) was used as the temperature standard. The obtained thermogram exhibited a single endothermic peak at 309 °C. More recently, several authors have reported OA melting points in the range 306–313 °C [1,4]. The OA boiling temperature has been established at 553–554 °C at 760 mmHg (https://pubchem.ncbi.nlm.nih.gov/compound/10494, accessed on 18 November 2021).

### 2.3. Spectral Information

Tong et al. [1] have recorded the UV spectrum of 98% purity OA in absolute ethanol at 60 µg/mL, showing a unique narrow absorbance peak at 205 nm (Figure 3). More recently, Verma et al. [4] have informed the UV spectrum of OA extracted from Lantana camera with the mixture chloroform:MeOH (60:40, *v/v*), which exhibited an isolated absorbance maximum around 210 nm.

The 1H-NMR spectrum of OA shows several tertiary methyl groups at δ 0.81, 0.87, 0.93, 0.94, 1.10, 1.15, and 1.20, and a characteristic olefinic proton of C12–C13 double-bonded pentacyclic triterpenoid at δ5.43 (1H, brs, H-12). This suggests an olea-12-ene skeleton. One methine proton at δ3.37 (1H, t, J = 8.2 Hz, 3α-H) is compatible with at least one hydroxyl group on the OA olean-12-ene-skeleton.

On the other hand, the 13C-NMR spectrum displays signals related to an oxygenated carbon signal at δ79.92 (C-3), one tri-substituted double bond at δ124.22 (C-12) and 146.47 (C-13) and one carboxyl group at δ181.86 (C-28). Further, 13C-NMR signals from C-18 to C-22 at δ(43.71 (C-18), 48.39 (C-19), 31.56 (C-20), 34.90 (C-21) and 32.57 (C-22)) indicates that OA derives from the oleanyl carbocation.

Hayashi and Tanaka (Hayashi T, Tanaka K. Oleonolic acid; LC-ESI-ITTOF; MS; [M-H]-. MassBank Accession: TY000153. Created 2010.10.14, modified 2011.05.06) have shown the MS spectra for OA obtained by LC-ESI-ITTOF, finding that the five most prominent peaks corresponded to the m/z molecular fragments (in decreasing order) 455.35; 456.35; 911.72; 912.70; 457.36. On the other hand, Guinda et al. [5] characterized OA by gas chromatography (GC) coupled with mass spectrometry (GC-MS), reporting an OA electron impact (EI) spectra recorded at 70 eV with the following five major m/z fragments: 203.2; 202.2; 189.2; 73.1; 187.2 (Figure 4).

## 3. Oleanolic Acid Extraction

Because pharmacologically active compounds are usually present in plants in low concentrations, an optimal extraction is essential for their recovery and purification from raw materials. An effective extraction strongly depends on their solubility and capacity of mass transfer through specific chemical environments [6]. OA and related triterpene are poorly soluble in aqueous mediums and many common organic solvents such as n-hexane [7], and their removal from plant tissues are usually subject to undesirable saturation by other co-extracted metabolites [8].

Due to the fact that both OA and related triterpenes are slightly polar compounds, medium-polar solvents should be selected. OA is highly soluble in solvents with Hildebrand solubility parameter (δ) in the range of 10–12 [9]. The best performance was obtained with n-butanol, which exhibits a Hildebrand’s δ of 10.4, very close to that reported for OA (δ 10.2) [7]. High yields are also achieved with ethyl ether, chloroform, methanol and ethanol. On the other hand, common nonpolar solvents such as toluene or cyclohexane gave poor extraction yields.

The sequential extraction of plant materials with petroleum ether followed by ethyl ether has been widely used for the recovery of triterpenoids, despite the possibility of eventually not exhausting the targeted compounds from the matrix. The former solvent concentrates the linear alkanes, fatty acids and alcohols and the latter one extracts triterpene acids, among other compounds [6]. Ethyl ether is slightly more effective than methanol and ethanol at solubilizing pure OA; however, these latter solvents produce a more exhaustive extraction due to their higher ability to percolate the plant matrix [6]. The polarities of ethanol and methanol match well to that of OA, resulting in high yields. The efficiency of the OA extraction by ethanol and methanol are similar [10]; however, methanol is a more toxic solvent than ethanol for humans and the environment, and therefore ethanol is the most commonly used extraction agent. Due to its low toxicity, ethanol is being preferred for practical use in the food, cosmetic and pharmaceutical industries. Different studies have revealed that aqueous ethanol is a better solvent than absolute ethanol for the extraction of triterpene acids from different natural products [11]. The ethanol diluted with water increased the yield versus pure ethanol because the water content increased the swelling effect of the plant tissue matrix, decreased the viscosity of the solvent, and improved the mass transport from the material, facilitating the extraction of triterpenes. The efficiency of the OA extraction reached maximum values at the range 70–95% ethanol using liquid-to-material ratios at the range 20:1–25:1 [3]. Utilizing more than 50% water in aqueous ethanol increased the polarity of the mixed solvent beyond the point at which it was suitable for extracting OA, and therefore, the yield decreased.

To solubilize OA in aqueous buffers, it should be first dissolved in water-miscible solvents such as dimethyl sulfoxide (DMSO) or dimethyl formamide (DMF) and then diluted in the aqueous buffers of choice. For instance, OA has a solubility of approximately 0.3 mg/mL in a 1:2 solution of DMF:PBS (pH 7.2) using this procedure.

The temperature is another factor affecting the recovery of triterpenes by liquid–solid extraction [12]. Warm temperatures can increase the diffusivity of the solvent into cells and enhance the desorption and solubilization of target compounds [13]. Xia et al. [10] studied the effect of temperature on the yields of triterpenes from *Ligustrum lucidum* by heat-reflux extraction, found that the yields of OA and ursolic acid (UA) were almost unchanged from 40 to 70 °C. The extraction of these triterpenes reached an equilibrium of desorption and solubility at 40 °C, and they were stable thermal components from 40 to 70 °C.

Different techniques have been implemented for the OA extraction from plant matrices. Conventional methods, such as maceration, heat reflux or Soxhlet, operate through cell permeation followed by solubilizing the active constituents by the extracting solvent [3]. They are often solvent- and time-consuming methods. Moreover, low selectivity and degradation of temperature-sensitive compounds may occur when using heat reflux or Soxhlet extractions [14].

Albi et al. [3] developed a procedure for extracting OA from the olive leaf by maceration with 95% ethanol at room temperature. The authors reported that the method was able to recover in a single extraction stage 90% of the OA leaf content. This method was also successfully applied for the extraction of triterpenes from the olive fruit [15] and the flesh and oil from *Argania spinosa* [16]. Similarly, a solid/liquid extraction by maceration was used to obtain ursolic and oleanolic acid from the apple cuticular layer [17]. The procedure comprised the first immersion in hot light petroleum ether to defeat the fruit skin and a consecutive immersion in hot chloroform.

Regarding the heat-reflux extraction (HRE), Wei et al. [18] compared the extracting efficiency of a low polar solvent, such as chloroform, with that of ethanol, a mid-polar solvent, for the extraction of OA and UA from *Hyedotis diffusa*. The highest amounts of OA were obtained when 70–95% ethanol was used. The yield of OA using chloroform was slightly lower, although similar to that when absolute ethanol was used.

In the last decade, there has been an increasing demand for novel extraction techniques, within the “green concept” (UN-2030 strategy), amenable to automation, with shortened extraction time, maximum efficiency and with a reduced requirement of organic solvents [19,20]. Alternative extraction techniques, such as subcritical water extraction [21], ultrasonic-assisted extraction (UAE) [22], solid-phase microextraction [23], microwave-assisted extraction (MAE) [24], pressurized liquid extraction [25] and supercritical fluid extraction (SC-CO_2_) [26], have raised the efficiency of the extraction process and diminished the number of organic solvents, operating temperatures and procedure times; however, each extraction technique presents advantages and disadvantages.

### 3.1. Ultrasound-Assisted Extraction of OA

The ultrasound-assisted extraction (UAE) has been successfully implemented for the extraction of triterpenes with satisfactory selectivity [27]. This method may enhance the extraction efficiency by accelerating the establishment of the equilibrium for OA dissolution between the plant cell wall and the extraction solvent [13]. Several probable mechanisms have been cited as the major factors for the UAE, such as cell wall disruption, particle size reduction, improved swelling and hydration processes and enhanced mass transfer of the cell contents as a result of cavitation bubble collapse [28]; therefore, UAE is an efficient and reduced solvent- and time-consuming technique, simple, economical and easily scaled up to the industrial level [29]; however, it should be considered that OA could be degraded by a long exposure to ultrasonic irradiation. Thus, 10 min could be considered as a safe time for OA extraction by UAE [30].

A series of modified ultrasound-assisted extraction (MUAE) methods coupling UAE with other conventional techniques have been developed, since they have been demonstrated to be good options for increasing the overall efficiency of the extraction operation [22]. Yang et al. [31] developed a hyphenated HRE-UAE process for extracting OA and UA from *Scutellaria barbata* D.Don, which consisted of an initial short-time HRE with stirring, followed by UAE without stirring. Optimal extraction was obtained with a particle size of 0.21–0.50 mm; ethanol concentration of 60%; ratio of solvent to raw material of 12:1 (mL/g); extraction temperature 55 °C; 5 min HRE and 20 min UAE at an ultrasonic frequency of 40 kHz and 180 W of power, duty cycle of 75% (intermittent pulse: 90 s on/30 s off).

More recently, ionic liquid-based ultrasonic-assisted extraction (ILUAE), combining the advantages of UAE and beneficial properties of IL, has been implemented. Ionic liquids (IL) are low melting point salts entirely ionized, with good solubility for organic compounds and miscibility with water, which interact with analytes through anion exchange, hydrogen bonding and hydrophobic interaction [32]. A large number of possible variations in cation and anion features allows the fine-tuning of the IL properties [33]. Zhang et al. [20] applied ILUAE to extract OA from grape seeds. A 0.7 mol/L 1-butyl-3-methylimidazolium chloride ((C4mim)Cl) solution in water was selected as the extractant. The optimal conditions of ILUAE were established as follows: extraction time 4 h; liquid–solid ratio 15:1 mL/g; ultrasonic power of 195 W; ultrasonication time 13 min; extraction temperature 48 °C. Compared with the regular UAE, the proposed ILUAE approach exhibited 1.64-folds higher efficiency.

### 3.2. Microwave-Assisted Extraction of OA

Microwave-assisted extraction (MAE) is another green technology with advantages over conventional methods for the extraction of secondary plant metabolites. MAE generates non-ionizing electromagnetic radiations, which rapidly increase the temperature of the dispersion medium by dipole rotation and ionic conduction [4,34,35]. Ionic conduction derives from the electrophoretic movement of ions produced by an electric field of microwaves, whereas dipole rotation involves collision among dipolar molecules. Both processes result in the generation of heat. Microwave energy is rapidly absorbed by polar substances in the cell matrix, drastically increasing its temperature and allowing the intracellular vaporization of liquids and the subsequent rupture of the cell wall. This cell lysis facilitates the mass transfer of solvent into the cell and phytochemicals into the solvent [36]. Major advantages of MAE are short extraction time, low energy requirement, high extraction efficiency and low degradation of target components [37]. On the other hand, MAE suffers from a few limitations. It is difficult to perform on entirely dry and too damp samples when a nonpolar solvent is used as an extractant. Furthermore, the productivity of MAE can diminish when the target compounds and/or solvents are non-polar. Finally, this experimental approach is incompetent when the viscosity of the extraction solvent is very high [37,38].

Sánchez-Ávila et al. [39] used MAE for the extraction of pentacyclic triterpenoids from the olive leaf. They concluded that the best results were obtained with 80% ethanol (8 mL/g plant material) and 180 W irradiation power. These authors realized a kinetics study to determine the time required for the total removal of the target compounds from the olive leaves, which was obtained after 5 min irradiation. The extracts obtained with longer times provided similar results with non-detected degradation.

Verma et al. [4] reported an MAE procedure for the rapid extraction and isolation of OA from roots of *Lantana camara* L. They obtained a maximum yield of 1.23% (DW roots), using a mixture of CHCl3:MeOH (60:40, *v/v*) as a solvent, and applying 600 W microwave power for 6 min at 50 °C. No degradation of the target analyte was observed at the optimum conditions, as evidenced by the recovery studies performed with standard OA. The proposed method also showed a high degree of reproducibility.

MAE was also implemented to extract triterpenoids (OA, UA, BA and Lupeol) from five *Swertia* species used as constituents in various herbal formulations [19]. The best MAE conditions for extraction were determined as 700 W microwave power, 2 min and 2 extraction cycles and by using aqueous ethanol (50%) at a ratio of 20 mL solvent per gram of plant material. Methanol and ethyl acetate were also used as extractants, although they offer worse results. In this investigation, MAE was demonstrated to be more efficient than conventional methods such as HRE or Soxhlet.

More recently, Tian et al. [40] have performed a comparative study for the extraction of triterpene saponins from *Aralia elata* (Miq.) Seem fruits and rachises, by using MAE, UAE, HRE and Soxhlet extraction. These techniques were compared from three perspectives: extraction kinetics, saponin yield and disintegration to the raw materials. The MAE parameters were optimized by the response surface methodology, finding maximal extraction with 92% and 91% ethanol, solid-to-liquid ratios of 1:30 and 1:22, microwave powers of 530 W and 400 W, irradiation times of 40 s and 54 s, for fruits and rachises, respectively. MAE had the highest extraction efficiency, since it required lower power and less extraction time to achieve the greatest disintegration of the raw materials, in comparison with the other extraction methods.

### 3.3. Supercritical Carbon Dioxide Extraction of OA

As supercritical carbon dioxide (SC-CO_2_) can often completely or partially substitute the use of the harsh chemicals required for conventional solvent extraction methods, it emerges as a promising technology for the effective extraction of high-value natural bioactives for human consumption. SC-CO_2_ is a green and environmentally friendly apolar solvent. It is able to dissolve lipophilic molecules from several plant matrices, so it is gaining a foothold in the industrial production of solvent-free vegetable oils. Further, it has been applied to concentrate lipophilic micronutrients in oil products [41]. Carbon dioxyde beyond its critical point (7.38 MPa and 31.1 °C) has been recognized as an excellent extraction solvent. The most important advantages of SC-CO_2_ include the preservation of thermolabile compounds, shortened extraction times, selectivity, and the ease of solvent removal from the extracted materials [42]. SC-CO_2_ has been successfully applied in the extraction of triterpenes from different vegetable matrixes at pressures in the range of 10–40 MPa and temperatures of 40–60 °C [43].

Pure SC–CO_2_ is not selective for OA and UA, and generated extraction yields similar to those obtained when *n*-hexane is used during HRE; however, adding an aqueous ethanol cosolvent can greatly improve the extraction efficiency due to the enhanced solubility of OA and UA after increasing the polarity of the SC–CO_2_. Furthermore, aqueous ethanol cosolvent accelerates the desorption process by reducing the interactions between the solutes and the sample matrix, competing with the solutes for an active binding site and disrupting the matrix structure. Performing SC-CO_2_ extraction, the yield of OA and UA increased significantly when the water (*v/v*) content of the aqueous ethanol increased from 0 to 18% (*v/v*) [18].

Domingues et al. [44] implemented a factorial design of experiments and response surface methodology to optimize the experimental conditions that maximize the extraction of triterpenes from the Eucalyptus globulus deciduous bark using SC-CO_2_. The triterpenoids fraction is mainly composed of pentacyclic triterpenic acids (TTAs), namely, betulonic, betulinic, 3-acetylbetulinic, ursolic, 3-acetylursolic, oleanolic and 3-acetyloleanolic acids. The best conditions were 200 bar, 40 degrees C and 5% ethanol, for which the statistically validated regression models provided: extraction yield of 1.2% (wt.), TTAs concentration of 50%, which corresponds to TTAs yield of 5.1 g/kg of bark and a recovery of 79.2% in comparison to the Soxhlet value. The behavior of the free and acetylated TTAs extraction was significantly different. The acetyl derivatives were extensively removed and tended to a plateau near 200 bar and 5% EtOH, while the free acids extraction always increased in the range of conditions studied. These results may be attributed to the inferior polarity of acetylated TTAs, and thus higher affinity to SC-CO_2_, when compared to the unesterified triterpenoids.

On the other hand, Wei et al. (2015) [18] developed a hyphenated process (HSC–CO_2_) consisting of ultrasound-assisted static stage, followed by SC–CO_2_ dynamic extraction (without ultrasound), that was employed for the obtaining of OA and UA from *Hedyotis diffusa*. This procedure involves a semicontinuous flow, high-pressure system. The herb sample was thoroughly mixed with stainless steel balls before being placed into the extraction vessel, which was then first submitted to ultrasonic treatment (15 min; frequency 40 KHz; duty cycle 79%) to allow contact between the samples and the supercritical solvent, which was followed by a dynamic extraction (without ultrasound-assisted) for 110 min, at a pressure of 28.2 MPa. The extractant was provided at a liquid/solid ratio of 50.6 mL/g and 80% aqueous ethanol (12.5% in SC-CO_2_) was used as cosolvent. At these conditions, the OA and UA yields were determined as 0.917 and 3.540 mg/g, respectively.

## 4. Quantitative Analysis Techniques for OA

Traditionally, the determination of OA and related triterpenes has been usually performed by GC with flame ionization detection (GC-FID) [45]. Because of their high molecular weight and low volatility, a previous derivatization step is mandatory [45]. The silylating reaction is a common derivatization step in (GC) analysis, in which the active (polar) hydrogen atoms in the analyte are replaced for a trimethyl-silyl-moiety to decrease the analyte boiling point and improve the chromatographic separation. Jemmali et al. [46] have optimized this reaction and reported that derivatization with N,O-bis(trimethylsilyl)trifluoroacetamide (BSTFA) and trimethylchlorosilane (TMCS) in pyridine (22:13:65 *v/v/v*) for 2 h at 30 °C is the most efficient method of derivatizing all the hydroxyl and carboxylic acid groups contained in the triterpene structures. It is notorious that the silylating reaction is not straightforward and is highly dependent on the nature of the derivatization reagent and solvent used. In addition, kinetics and the yield of the reaction can be dependent on the structure of the compound and the number and type of functional groups to be derivatized.

For the GC separation of triterpenic acids, both nonpolar and mid-polar fused silica capillary column with methyl phenyl polysiloxane bonded stationary phase have been used [45], with hydrogen or helium as carrier gases and oven temperatures in the range 240–300 °C, operating in isothermal or gradient program.

In the last two decades, a number of different analytical techniques have been used to quantify OA in plant extracts, including high-performance liquid chromatography with photodiode-array detection (HPLC-DAD) [28], gas chromatography coupled with mass spectrometry (GC-MS) [47], high-performance liquid chromatography coupled with mass spectrometry (HPLC-MS) [48], thin-layer chromatography (TLC) [49], supercritical fluid chromatography (SFC) [50], nuclear magnetic resonance spectroscopy (NMR) [51], micellar electrokinetic chromatography (MEKC) [52], capillary zone electrophoresis [53] and capillary supercritical fluid chromatography [54].

Despite GC exhibiting higher sensitivity, HPLC-DAD is currently the most extended method for quantitative analysis of OA and related plant triterpenes. Almost all of the HPLC-DAD protocols described in the literature make use of reverse phase C18 chromatographic columns thermostated in the range 20–27 °C, record UV spectra between 190 and 400 nm for peak characterization and set the detection wavelength at 210 nm. The differences between the procedures derive from the type of eluents selected and whether they run in isocratic or gradient elution [10]. The low UV absorption provided by the saturated skeleton of pentacyclic triterpenes could underlay the aforementioned limited [55,56]. To overcome these shortcomings, liquid chromatography coupled to mass spectrometry detection (LC–MS) has become a powerful hyphenated technique that enables the separation, unambiguous detection and characterization of bioactive compounds in complex samples.

## 5. Oleanolic Acid Biosynthesis

Higher plants produce a vast array of non-steroidal triterpenes. More than a hundred different carbon skeletons have been described, and further oxidative and glycosative modifications generate an even greater structural diversity [57,58]. Pentacyclic triterpenoids and phytosterols are synthesized via the cytoplasmic acetate/mevalonate pathway and share common biosynthetic precursors up to (3S)-2,3-squalene epoxide, also known as oxidosqualene [59,60]. From here, the diversity of resulting triterpenic structures is controlled by the stereochemical conformation of the substrate imposed by the enzymes catalyzing the reaction, called oxidosqualene cyclases or triterpene synthases [61]. Cycloartenol synthase cyclizes oxidosqualene folded in the chair–boat–chair conformation via the protosteryl cation into cycloartenol, the first cyclic precursor of the sterol pathway, whereas non-steroidal triterpenoids are assumed to be formed from oxidosqualene folded in the all-chair conformation. The cyclization of oxidosqualene constitutes a branching point between the primary metabolism (sterol pathway) and the secondary one (triterpenoid biosynthesis) [62].

A number of triterpene synthase cDNAs have been cloned from plant sources and their enzyme functions identified by heterologous expression in yeast [63]. These studies have shown that most triterpene synthases are multifunctional and are able to simultaneously form a vast array of products from oxidosqualene [64]. By generating four or five rings and several asymmetric centers in a single step, the biosynthesis of the non-steroidal triterpene is thought to proceed through a series of rigidly held carbocationic intermediates [65]. The tetracyclic dammarenyl C-20 cation is first produced, and its subsequent rearrangement leads to different pentacyclic cationic intermediate, such as baccharenyl, lupenyl, oleanyl and ursanyl cations. The formed pentacyclic triterpenes are then metabolized into more-oxygenated compounds, which may remain as free alcohols and acids or be conjugated as glycosides (triterpenic saponins) and fatty acids esters [66]. For instance, in *Oleae europaea*, oxidosqualene cyclization by β-amyrin synthase yields β-amyrin, which by further three consecutive oxidative steps at the C-28 position, catalyzed by the cytochrome P450 (CYP) enzyme, produces erythrodiol, oleanolic (OA) acid and maslinic acid (MA).

## 6. Food Sources of OA

Oleanolic and ursolic acids (UA) are two of the most common pentacyclic triterpenoids in nature. They are ubiquitous in the plant kingdom, where they usually occur in both their free form and as a triterpenoid saponin aglycone linked to one or more sugar moieties. In their free form, they are found in plant cuticular waxes, where they participate in the plant response against biotic and abiotic stresses; therefore, pentacyclic triterpenic acids are natural components of many foods and medicinal herbs and an integral part of the human diet.

A number of spices and fruits contain, besides other nutraceuticals, pentacyclic triterpenes from the lupane, oleanane and ursane groups. For example, they can be found in rosemary and other spices of the Lamiaceae family as well as within olive leaves and fruit [67]. Virgin olive oil contains up to 197 mg/kg triterpenes, indicating the importance of these substances as nutraceuticals [68]. Apples are among the most consumed fruits worldwide and their anti-tumoral effects have been attributed to components of the peel [69], including OA, UA and maslinic acid (MA) [70].

## 7. Biological Activity of OA

Oleanolic acid and related triterpenes are endowed with a wide range of pharmacological properties whose therapeutic potential has only partly been exploited until now. Throughout complex and multifactorial mechanisms, they exert beneficial effects against dyslipidemia, diabetes and metabolic syndrome. They improve insulin response, preserve functionality and survival of β-cells and protect against diabetes complications. There is evidence of an antiviral, anti-HIV, antibacterial, antifungal, anticarcinogenic, antidiabetic, anti-inflammatory, hepatoprotective, gastroprotective, hypolipidemic and anti-atherosclerotic effect [71,72,73,74]. In addition, pentacyclic triterpenoids interfere in several stages of the development of different types of cancer, inhibiting the genesis and evolution of the tumor and inducing apoptosis of tumor cells [75,76,77]. Most pentacyclic triterpenic acids have similar pharmacological properties, although they can differ in the intensity of biological activity due to their respective molecule features, which influence their potency and consequently their bioactivity.

### 7.1. Absorption, Distribution, Metabolism and Excretion (ADME) of OA

Due to their hydrophobic nature, pentacyclic triterpenoids are almost insoluble in water. They are classified in the biopharmaceutical classification system as class IV drugs as their pharmacological effects are limited due to their low solubility in water and potential difficulty in permeating biological membranes [78]. For these reasons, some new approaches have been introduced to enhance their biopharmaceutical properties, particularly drug delivery technologies. At present, nano-emulsions, mesoporous silica nanoparticles, solid lipid nanoparticles, liposomes, niossomal gels and solid dispersions have been proposed quite successfully [79]; however, most of these studies were performed in experimental animals, and research in humans is very scarce, despite the great therapeutic potential of these natural biocompounds.

Table 1 summarizes the current knowledge of OA pharmacokinetics in humans after oral single-dose administration. Song et al. [80] reported the results of a pharmacokinetic study performed with 18 healthy male Chinese volunteers, in which the participants received a single dose of 40 mg OA (capsule preparation). Venous blood samples were collected for 48 h, and the plasma was analyzed for OA contents. The area under the plasma OA concentration–time curve (AUC) was determined in 124.3 ± 106.7 ng h/mL, with a peak OA concentration of 12.1 ± 6.8 ng/mL, which was registered at 5.2 ± 2.9 h after the OA intake. Later on, a randomized, crossover and self-control study was carried out with 18 healthy Chinese male volunteers, who received 20 mg OA orally, formulated as normal tablets and dispersible tablets [81]. The resulting pharmacokinetic parameters included AUC values of 112.2 ± 56.6 and 109.7 ± 41.6 ng h/mL, for normal and dispersible tablets, respectively, and OA peak concentrations of 18.9 ± 8.0 and 17.8 ± 7.5 ng/mL, which were recorded at 2.9 ± 1.2 and 2.5 ± 1.0 h after the triterpene administration. In 2015, Rada et al. informed on a pharmacokinetic study developed with nine adult men, who were administered 30 mg OA dissolved in 70 g of pomace olive oil. The reported AUC value was 3181.9 ± 894.3 ng h/mL, with an OA maximum concentration of 598.2 ± 176.7 ng/mL, achieved at 3.0 ± 0.8 h after the intake of the OA-containing oil. The administered OA dose revealed a distribution volume of 81.4 ± 9.7 L, and removal half-lives of 4.6 ± 1.1 h. The 30 mg OA oral dose yielded a total clearance of 35.1 ± 4.2 L/h [82]. More recently, de la Torre et al. [83] reported that an administration of 4.7 mg of OA within 30 mL of olive oil to 12 individuals led to a maximum concentration of 5.1 ± 2.1 ng/mL in plasma 4 h after the intake.

Serum albumin is an important plasma protein responsible for the binding and transport of many endogenous and exogenous substances such as hormones and fatty acids, as well as foreign molecules, such as drugs. It is known that the distribution, free concentration and metabolism of various drugs are strongly affected by drug–protein interactions in the bloodstream. Many drugs and other bioactive small molecules bind reversibly to albumin and other serum components, which then function as carriers. In their study, Rada et al. [84] showed that 98.9 ± 2.5% of the circulating OA was bound to human serum proteins (mainly albumin). The thermodynamic analysis showed that the basic forces acting between bovine serum albumin and the triterpene were hydrogen bonds, van der Waals forces and hydrophobic interactions [85]. Dopierala et al. [86] found that the binding of OA to albumin can be irreversible according to pH, ionic strength and temperature, which may have a significant impact on the distribution of oleanolic acid in the human body. Peng et al. [85], Bhattacharya et al. [87] and Subramanyam et al. [88] anticipated that the complexation of plant triterpenoids with protein might be exploited as a biologically relevant model for evaluating the physiologically applicable no-covalent complexes in the in vivo examination of triterpenoid properties such as accumulation, bioavailability and distribution.

### 7.2. OA Effects on Lipid Profile and Obesity

#### 7.2.1. Animal Models

Since the 1990s, the hypolipidemic abilities of pentacyclic triterpenoids have been are well-established [89]. OA and MA diminished plasma concentrations of triglycerides (TG), total cholesterol (TC), HDL and LDL, and downregulated the expression of lipogenic genes (acetyl-CoA carboxylase (ACC), stearoyl-CoA desaturase 2 (SCD2), glycerol-3-phosphate acyltransferase (Gpam) and acyl-CoA cholesterol acyltransferase (ACAT)) in rats fed a high-cholesterol diet [90]. Further, OA and BA significantly decreased visceral fat in obese Swiss mice, with an augmentation of leptin and a reduction in plasma lipids and ghrelin. Triterpenes considerably lower microvesicular steatosis and lipid droplets in the liver induced by this diet [91]. The 18b-Glycyrrhetinic acid also reduced TG, TC, free fatty acids (FFA), phospholipids, LDL and VLDL in plasma, kidney, liver and heart of STZ diabetic rats [92].

A study conducted by Jiang et al. [93] with HFD-fed atherosclerotic male quails revealed that OA treatments (25–100 mg/kg/day, via gavage) produced a significant reduction in TG, TC and LDL, as well as a significant increase in HDL and nitric oxide (NO) serum levels. On the other hand, Pan et al. [94] have reported that in New Zealand, White rabbits fed an atherogenic fat diet (1% cholesterol and 5% lard oil) for 12 weeks (120 g/day), the administration of 50 mg OA/kg/d for 28 days reversed the high-fat-induced upregulation of TC, TG, LDL-C and HDL-C. The intraperitoneal administration of OA (20 mg/kg/day for two weeks) to Lep db/db obese diabetic mice caused a significant decrease in TG, TC, LDL and free fatty acids serum levels and a significant increase in HDL levels in comparison to the non-OA-treated diabetic animals [95]. OA treatment also reduced body weight, liver weight and fat weight, whereas it protected liver morphology and function. Similarly, Gamede et al. [96] have shown that OA significantly decreased body weight, TG, and LDL in plasma of prediabetic rats and significant rise of HDL. Further, OA administration lowers saturated FFA and increases mono/polyunsaturated FFA in neonatal rats fed with an HF diet [97], and significantly diminishes the plasma levels of octanoylated ghrelin and the body weight gain in comparison to non-OA-fed rats [98]. In the liver of spontaneously hypertensive rats (SHR) treated with OA (1.08 mg/kg) for 4 weeks, Zhang et al. [99] observed a decrease in TG levels; however, they also found increased plasma LDL-C levels, with no significant effects on HDL-C and total cholesterol.

Chen et al. [100] studied potential hepatic targets of the lipid-lowering effect by OA in hyperlipidemic mice, finding that acute as well as chronic OA treatments (20 mg/Kg BW) reduced levels of TG, TC and LDL in serum, and the expression of peroxisome proliferator-activated receptor-γ coactivator-1b (PGC-1b) and its downstream target genes in the liver, pivotal in maintaining lipid homeostasis. OA accelerated mRNA degradation of PGC-1b, likely via regulation of the miR-98–5p/PGC-1b axis. Furthermore, Luo et al. [101] investigated the influence of OA on serum lipid levels and lipid accumulation in the liver, using three animal models: C57BL/6J mice, LDL receptor knockout (LDLR−/−) mice and rabbit that mimicked atherosclerosis. In the three models, OA significantly lowered the plasma levels of TG, TC and LDL. In addition, in the LDLR−/− mouse model, OA enhanced serum HDL. The authors provided evidence that these effects may be derived from a regulatory role for OA on the expression of genes involved in lipid metabolism, such as peroxisome proliferator–activated receptors-γ (PPARγ), AdipoR1 and AdipoR2. Indeed, in rabbit model, the mRNA levels of AdipoR1 were markedly increased, while those of AdipoR2 was remarkably decreased by the OA treatment. In the LDLR−/− mouse model, OA elevated the levels of AdipoR1 and PPARγ.

#### 7.2.2. Transcriptional Gene Expression

Lipid metabolism and glucose homeostasis are regulated by PPARs, as transcriptional regulators of genes involved in these pathways. OA and related triterpenoids may modify PPAR activity. As a PPAR-α agonist, OA improved the cardiac lipid metabolism of ZDF diabetic rats [102] and activated the differentiation of keratinocytes HaCaT and renal fibroblast (CV-1) cells [103]. Further, the transactivation of PPAR-γ by OA activates a hypoglycemic effect in diabetic KK-Ay mice [104]. In contrast, PPAR-γ downregulation is induced by reduced lipid accumulation and visfatin levels in differentiated 3T3-L1 adipocytes [105]. Two glycosylated OA derivatives from *Kalopanax pictus* are able to transactivate the three PPAR subtypes [106], which is very promising for the treatment of metabolic diseases as they could simultaneously target several related signs, such as insulin resistance, atherogenic dyslipemia and obesity/overweight.

#### 7.2.3. Effects in Humans

In humans, a proof-of-concept clinical trial, conducted by Luo et al. with 15 hyperlipidemic patients, showed that the administration of OA for four weeks caused a decrease in TC, TG, LDL, glucose and insulin serum levels, as well as an increase in leptin serum levels. A slight decrease in HbA1c (%) and a slight increase in HDL were also observed [107].

### 7.3. OA Effects on Carbohydrate Metabolism

Digestion of complex carbohydrates is facilitated by enteric enzymes, such as α-glucosidase and α-amilase, in the brush border of the small intestine cells. Their inhibition allows better control of postprandial hyperglycemia and originates, in the long term, a modest reduction in glycosylated proteins. Pentacyclic triterpenes have been revealed as potent inhibitors of α-glucosidase in vitro, in an uncompetitive and dose-dependent fashion (half-maximal inhibitory concentration (IC50) 10–15 mmol/L) [108]. Wang et al. [56] reported that OA presents a high affinity for α-glucosidase, with a calculated dissociation constant (Kd) of 44.9 μM.

OA also inhibits the pancreatic and salivary α-amilase activity (IC50 0.1 mg/mL), eliciting a hypoglycemic effect in prediabetic patients (impaired fasting glucose) fed cooked rice. At a dose of 1 mg/kg, OA reduced blood glucose by 23%, 30 min after the meal [109]. A similar hypoglycemic effect was observed in diabetic GK/Jcl rats fed starch [109].

In type 2 diabetes, pancreatic β-cells fail to release enough insulin to compensate for hyperglycemia. This deficit involves morphological and functional β-cell alterations. Accumulated data indicate that pentacyclic triterpenoids increase biosynthesis and secretion of insulin and improve glucose tolerance through a multifactorial mechanism.

#### 7.3.1. Activation of the β-Cell M3 Muscarinic Receptors

In Wistar rats, the intraperitoneal injection of OA reduced fasting glycemia in parallel with the increase in plasma insulin [110]. These actions were abolished by hemicholinium-3 and vesamicol, inhibitors of the choline uptake and acetylcholine transport, respectively, which suggests that OA might raise acetylcholine release from nerve terminals and facilitate the glucose-dependent insulin release by activating the M3-subtype muscarinic receptors in the pancreatic β-cell membrane [111].

#### 7.3.2. Agonist Action on the TGR5 Receptor

Bile acids emerged as signaling molecules with systemic endocrine functions, which induce the production of glucagon-like peptide-1 (GLP-1) throughout a TGR5 (Takeda G protein-coupled receptor 5)-mediated mechanism that increases insulin secretion and β-cell regeneration [112]. OA performs as a selective TGR5 agonist (EC50 1.42 mmol/L) [113], since at difference of bile acids does not activate the FXR receptor. TGR5 stimulation also increases cAMP production and the thyroid hormone activating enzyme deiodinase type 2 activity in brown adipose tissue, enhances oxidative phosphorylation in muscle and stimulates eNOS expression and improves the immune and inflammatory responses in enteroendocrine cells [114]. Maczewsky et al. [115] reported that OA directly stimulates β-cells by binding to the TGR5, increasing the cytosolic Ca2+ concentration by stimulating Ca2+ influx. They also identified Protein kinase A (PKA) as a downstream target of TGR5 activation. In the liver, the effect of OA on TGR5 differs according to the dose. At low doses, it induces hepatoprotection, but at high doses, it could be hepatotoxic [116].

#### 7.3.3. Enhancement of the Shp-2 Enzyme Activity

The Src homology phosphotyrosyl phosphatase 2 (Shp-2) is involved in receptor-activated pathways, such as insulin biosynthesis and signaling. Cytoplasmic Shp-2 plays a pivotal role in the transcription of the insulin gene, adjusting signals through PI3K/Akt/FoxO1 and extracellular signal-related kinase (ERK) pathways that finish controlling homeobox 1 (Pdx1) gene expression in the pancreas and duodenum, and the activity on Ins1 and Ins2 promoters. OA enhances the Shp-2, promoting a hypoglycemic effect in streptozocin (STZ) diabetic mice in a dose-dependent manner [117]. It is a selective activity as it does not trigger other phosphotyrosyl phosphatases, such as Shp1, Vhr and HePTP. The chronic administration of OA (30–50 mmol/L) activated insulin biosynthesis at the transcriptional level in INS-1 rat β-cells, enhancing the insulin protein level (~25%) and increasing proinsulin and insulin-2 mRNA [118]. Moreover, OA enhanced mitochondrial Shp-2 activity and blocked the caspase-3 apoptotic pathway [119].

#### 7.3.4. Promotion of β-Cell Survival and Proliferation

β-Cells are notably sensitive to cytokines, which activate islet degeneration and cell death [120]. OA extended survival of transplanted islets in STZ diabetic mice by inhibiting the cytokine production by macrophages and antigen-presenting and other infiltrating cells [121]. OA significantly diminished serum IP-10 and interleukin (IL)-4 cytokines and lowered the frequency of γ-interferon-, IL-4-, IL-7-, and IL-2-producing T cells.

### 7.4. Improvement of the Insulin Signaling

Consistent evidence indicates that pentacyclic triterpenoids stimulate the activation of the insulin receptor (IR) and downstream signaling pathway. OA enhanced mRNA expression of the genes encoding insulin receptor, insulin receptor substrate (IRS)-1 and phosphatidylinositol 3-kinase in white adipose tissue of rats. OA also upregulated total IRS-1 expression, eliminated the increased phosphorylated IRS-1 at serine-307 and fixed up the increased phosphorylated IRS-1 to total IRS-1 ratio at the protein level. On the contrary, the phosphorylated-Akt/total-Akt ratio was augmented; therefore, it was suggested that OA supplement could be beneficial in fructose-induced Adipo-IR via the IRS-1/phosphatidylinositol 3-kinase/Akt pathway in rats.

#### 7.4.1. Insulin-Mimetic Effect as IR Co-Activator

OA and related triterpenes act as activators of the IR that synergistically enhance the low-dose insulin-mediated IR autophosphorylation [122]. In the absence of the hormone, low doses of the triterpenoids (1 mg/mL) could not activate IR, although at concentrations higher than 50 mg/mL, they duplicated the number of autophosphorylated receptors.

#### 7.4.2. Inhibition of Protein-Tyrosine Phosphatases PTP1B and TCPTP

Tyrosine phosphatases PTP1B and TCPTP negatively regulate insulin signaling in vivo. Inhibition of these proteins improves insulin sensitivity and stimulates glucose uptake [123]. Oleanane- and ursane-type triterpenes inhibit PTP1B in a potent, selective and reversible way. The mechanism could be referred to as linear-mixed type [124]. OA and UA discriminate other phosphatases involved in the insulin pathway but do not exhibit obvious selectivity among PTP1B and TCPTP [125].

#### 7.4.3. Activation of PI3K/Akt

PKB/Akt is essential for insulin-stimulated events such as glucose uptake and glycogen synthesis. OA and 3β-taraxerol have been reported for stimulating Akt in vascular smooth muscle cells and 3T3-L1 adipocytes, respectively. Because wortmannin blocked these events, the requirement of PI3K activation seems obligatory [126]. The involvement of the PI3K/Akt pathway in the pharmacological effects of OA has also been observed in the improvement of symptoms of renal ischemia reperfusion injury [127], allergic responses [128] and apoptosis and autophagy of SMMC-7721 hepatoma cells [129].

It has also been described that OA exerts a potent glucose-lowering effect in diabetic mice that is sustained well beyond the treatment period [130]. The downregulation of glucose-6-phosphatase, the gatekeeping gluconeogenesis enzyme, induced by AKT and FOXO1, was proposed as the likely mechanism.

#### 7.4.4. Activation of LKB1/AMPK

AMP kinase (AMPK) participates in several metabolic pathways, including muscular glucose uptake and fatty acid oxidation and hepatic fatty acid synthesis and gluconeogenesis. AMPK is activated when the ATP/AMP ratio diminishes as an answer to fasting and pathological stresses. OA, UA and BA stimulate AMPK in human HepG2 hepatocytes [131]. Similarly, in KK-Ay mice, MA improves insulin sensitization [132], and experiments with the synthetic OA derivative 2-cyano-3,12-dioxooleana-1,9-dien-28-oic acid methyl ester (CDDO-Me) have provided evidence that this event is mediated by phosphorylation of both hepatic kinase B1 (LKB1) and AMPK. The concrete step triggering LKB1/AMPK remains unclear, although it seems that triterpenes target the upstream kinase ERK1/2 [133]. In neonatal rats fed a high-fructose diet, OA increased AMPK gene expression in skeletal muscle by 4-fold compared to control, together with GLUT-4 [134]. This was observed concomitantly to a reduction in the high plasma levels of TNF-α and IL-6 caused by the high-fructose diet, as well as to an increase in adiponectin.

#### 7.4.5. Inhibition of Glycogen Synthase Kinase-3β

Glycogen synthase kinase (GSK) 3β reduces the insulin signal without stimulus. OA, UA and 3β-taraxerol seem to have the ability to stimulate glucose uptake and glycogen synthesis by means of the inhibition of GSK3β [135]. Other studies have indicated that triterpenes, such as other GSK3β inhibitors, attach to the ATP binding site by hydrogen bounds [136].

#### 7.4.6. Effects on the Glycogen Pool

Diabetes lowers glycogenesis and increases glycogenolysis. Pentacyclic triterpenoids reversed these disorders. Indeed, UA increases glycogen in the liver of STZ diabetic mice by stimulating glucokinase and inhibiting glucose-6-phosphatase [137]. Further, OA and other structural analogs repressed glycogen phosphorylase, the rate-limiting step of glycogenolysis, in adenocarcinomic A549 cells (IC50 5.98 mmol/L) [138] and rabbit muscle (IC50 14 mmol/L) [139], with triterpenes binding at the AMP allosteric site in the enzyme [140].

### 7.5. Effects on Metabolic Syndrome Components

Metabolic syndrome consists of a cluster of pathophysiological conditions risk factors, including obesity, dyslipemia, hypertension and hyperglycemia that lead to type 2 diabetes mellitus and cardiovascular disease [141,142]. In addition, metabolic syndrome is also associated with oxidative stress, hepatic steatosis, non-alcoholic fatty liver disease and impaired glucose tolerance, which can be the subject of the OA action [143].

#### 7.5.1. Hypotensive Effect

Different studies performed with hypertensive animals have demonstrated that OA may reduce both systolic blood pressure (SBP) and mean arterial pressure (MAP) and increase the urinary excretion of sodium. Indeed, Ahn et al. [144] reported that the application of 20–30 mg OA/kg/day over a period of three months produced a significant decrease in SBP in renovascular hypertensive rats in comparison to non-OA-treated animals. Similarly, the administration of 60 mg OA/Kg/day for four weeks to Nω-nitro-L-arginine methyl ester (L-NAME)-induced hypertensive rats significantly reduced SBP and MAP [145]. OA also increased urine volume and urine sodium excreted, as well as augmented non-significantly serum nitrate/nitrite (NOx) levels. Another study involving sub-chronic treatments (9 weeks) with OA and synthetic derivatives (30–120 mg/kg) obtained comparable results in SHR and Dahl salt-sensitive (DSS) rats [146]. These treatments significantly reduced MAP and increased urinary Na+ excretion. Likewise, Gamede et al. [96] have reported a reduction in the MAP in prediabetic male Sprague–Dawley rats. More recently, Zhang et al. [99] attributed the antihypertensive effect of OA to the regulation of liver lipid metabolites. They observed that the protein and mRNA levels of secretory phospholipase A2 (sPLA2) and fatty acid synthase (FAS) were reduced in SHR rats treated with OA. These enzymes are involved in the conversion of phosphatidylcholine into lyso-phosphatidylcholine and in de novo synthesis of palmitic acid, respectively.

#### 7.5.2. Improvement of Hyperglycemia and Insulin Resistance

Wang et al. [95] showed that administration of OA to diabetic mice significantly decreased body weight, fasting blood glucose (FBG) and fasting serum insulin (FSI). Li et al. [147] investigated the effect of OA on adipose tissue insulin resistance (Adipo-IR) since Adipo-IR contributes to the progress of metabolic abnormalities through the release of excessive free fatty acids from adipose tissue. Male Sprague–Dawley rats fed high-fructose (10%) drinking water were treated with OA (25 mg/kg, once daily) by oral gavage over 10 weeks. The triterpene treatment reversed the increase in the Adipo-IR index and plasma FFA during an OGTT. Furthermore, OA attenuated the fructose-induced increase in plasma insulin and the HOMA-IR score. Similarly, Otsuka Long-Evans Tokushima Fatty diabetic rats treated with OA (100 mg/kg/day) by oral gavage for 20 weeks experienced a significant increase in the HOMA for β-cell function (HOMA-β) and insulinemia [148]. Moreover, administration of OA slightly decreased glucose AUC during an IPGTT and an intravenous insulin tolerance test (IVITT). On the other hand, Male C57BL/KsJ-Lepdb (db/db) diabetic mice administered with OA (250 mg/kg) significantly reduced FBG, HOMA-IR and serum HDL levels and non-significantly reduced FSI [149]. More recently, Gamede et al. [150] conducted studies in high-fat high-carbohydrate (HFHC) diet-induced prediabetic male Sprague–Dawley rats, in which OA administration (80 mg/kg) for a period of 12 weeks originated a significant reduction in body weight, glycemia, HOMA2-IR and HbA1c indexes, as well as a reduction in the hepatic and muscle glycogen concentration in comparison to the non-OA-treated control rats. Likewise, Djeziri et al. [151] have pointed out that supplementation of drinking water with OA (50 µg/mL) for 16 weeks to HFD-induced obese C57BL/6J female mice, caused a signification reduction in glycemia after an IPGTT test. More recently, Claro-Cala et al. [152] administered a pomace olive oil rich in triterpenic compounds, including OA, to obese mice for 20 weeks and reported a significant improvement in oral glucose tolerance and intraperitoneal insulin sensitivity compared to obese control mice. The intake of the triterpenoid-rich oil improved both the oral glucose (GTT) and insulin tolerance tests (ITT). In fact, OA has been shown to coadjuvant metformin to ameliorate the consequences of pre-diabetes in Sprague–Dawley rats a high-fat high carbohydrate diet for 20 weeks [153]. According to these authors OA improved lipid metabolism, decreased plasma proinflammatory cytokines, as wells as C-reactive protein, fibrinogen and CD40L in the animals.

Exposure to polychlorinated biphenyls is closely associated with obesity and diabetes. In male C57B6/J mice, Aroclor-1254 induced adiposity and insulin resistance and resulted in a significant increase in FBG, insulin, HOMA-IR, TG, FFA and TC. Pre-treatment with OA (50 mg/kg) prevented these abnormalities and significantly attenuated the increase in adipose weight and adipocyte size and repressed adipocyte differentiation [154]. Furthermore, OA markedly inhibited the increase in the ROS production and NOX4 expression induced by Aroclor-1254. OA could inhibit PCB-induced oxidative injury and glucose/lipid metabolic dysfunction via hepatocyte nuclear factor 1b (HNF1b)-mediated regulation of redox homeostasis.

#### 7.5.3. Inhibition of Polyol Pathway and AGEs

Chronic hyperglycemia significantly raises the polyol pathway in insulin insensitive tissues. The rate-limiting aldose reductase (AR) reduces glucose to sorbitol, which is further metabolized by sorbitol dehydrogenase (SDH) to fructose. High levels of sorbitol and fructose activate the synthesis of advanced glycation end products (AGEs) and promote stress-sensitive signaling pathways. OA and UA inhibit AR, as well as SDH, in the liver and kidney of STZ diabetic mice [137]. In addition, OA enhances glyoxalase-I (GLI) and diminishes methylglyoxal concentration. In vitro, OA inhibits the formation of pentosidine and Nɛ-(carboxymethyl)lysine (CML) in a dose-dependent manner [155]. OA and UA also lowered the levels of plasma HbA1c, the renal pentosidine and CML and urinary glycated albumin, being this inhibitory potency greater for OA than that of UA at a similar concentration in alloxan diabetic mice [156].

#### 7.5.4. Alleviation of the Oxidative Stress-Induced Insulin Resistance

Experimental and clinical evidence suggests an inverse association between insulin sensitivity and oxidative stress. Dysfunctions in mitochondria and endoplasmic reticulum are related to an excessive ROS production or an impairment of the endogenous antioxidant system [157]. The antioxidant power of OA is controversial as it is markedly influenced by the experimental systems used to evaluate it. Anyway, it is generally accepted that OA and related triterpenoids exhibit a very moderate aptitude as radical scavengers [158]. Still, Gutierrez et al. [159] found that treatment with OA prevented lipid peroxidation and superoxide anion accumulation in the gut, while inducing the expression of the ROS scavenger Sestrin-3, which was associated with a normalization of the levels of intestinal mucosal dysfunction markers, as well as the pro- and anti-inflammatory immune bias in a murine model of experimental autoimmune encephalomyelitis.

Nevertheless, OA exhibits a rather efficient antioxidant defense activity in cell cultures and experimental animals. Recently, there is more evidence about OA and its analogs intensify the cellular antioxidant response by modulating the gene expression of antioxidant proteins and NADPH-producing enzymes, responsible for maintaining the GSH level and its transportation into mitochondria. In fact, OA upregulated the expression of glutathione peroxidase (GSHPx) and superoxide dismutase (SOD) in hepatocytes treated with tert-butyl hydroperoxide [160,161]. In addition, OA and UA improved the viability of PC12 cells treated with H2O2 or 1-methyl-4-phenylpyridinium by increasing glutathione content and catalase and SOD activities. They also decreased the lactate dehydrogenase release and malondialdehyde formation [162]. Moreover, OA enhances the glutathione-mediated mitochondrial antioxidant mechanism, protecting against myocardial ischemia reperfusion injury in rats [163]. OA also raised GSHPx and SOD activities and diminished malondialdehyde levels in the liver and kidney of alloxan diabetic rats [156]. In these effects, the activation of the nuclear factor erythroid 2 p45-related factor 2 (Nrf2) seems to play a pivotal role [164].

### 7.6. Oleanolic Acid and Gut Microbiota

Insulin resistance and hyperglycemia are the main underlying alterations behind metabolic syndrome. Insulin resistance may result in full-blown diabetes when pancreatic beta-cells become too damaged and incapable of supplying enough insulin [165]. Insulin resistance is associated with a state of chronic systemic low-grade inflammation [166]. Obesity, insulin resistance and low-grade inflammation are mainly triggered by an excess of circulating fatty acids from dietary fat [167]. Fat can also trigger systemic inflammation and insulin resistance via changes in the populations of gut microbiota (gut dysbiosis) [168,169], probably through lipopolysaccharide (LPS: a component of the cell wall of Gram-negative bacteria) induced inflammatory endotoxemia [170]. Other possible mediators linking gut bacteria to insulin resistance and adipose tissue function are short-chain fatty acids (SCFAs: end products of the fermentation of dietary fiber by bacteria) and bile acids [171].

Due to its low solubility and bioavailability, OA remains in the alimentary tract after their intake and it directly interacts with the gut microbiota of the host; however, the effects of oleanolic acid on gut microbiota have scarcely been studied. The growth of Lactobacillus is stimulated by OA in a dose-dependent manner up to 4% (*w/v*) in vitro [172], and this activity would be related to its anti-inflammatory effects. In a preclinical model, the family Muribaculaceae was recovered by treatment with OA (25 mg/kg/day, i.p, for 21 days) in the fecal microbiota of mice induced autoimmune myocarditis [173]. This effect was reported together with an improvement in several pro-inflammatory mediators, such as sPLA2-IIA and IL-1β, and certain protection from oxidative stress [173]. Moreover, bardoxolone methyl (BARD), a synthetic pentacyclic triterpenoid and derivative of oleanolic acid, prevented microbial population changes induced by a high-fat diet in mice [174]. Concretely, BARD intake balanced the proportions of Bacteroides/Prevotella, *Bifidobacterium* spp. and *Lactobacillus* spp. in cecal content [174]. Further, a study about *Ligustrum lucidum* (LL), a Chinese herb rich in OA, has demonstrated certain activity on cecal microbiota of laying hens supplemented with 2% LL [175]. Concretely, at the phylum level, the main differences were detected in the relative abundance of Bacteroidetes, Firmicutes, Saccharibacteria and Verrucomicrobia, while at the genus level, the main changes were at *Phascolarctobacterium*, *Bacteroidales*, *Butyricicoccus*, *Ruminococcaceae* and RC9 gut and coprostanoligenes groups.

Further studies are needed to fully determine the effects of OA in the gut microbiota. Meanwhile, other related triterpenes, such as ursolic acid (UA), can give clues about the effects of OA on the gut microbiota. Ursolic acid administered p.o. to rats alters the diversity of intestinal microbiota and enhances intestinal health by inhibiting the gut colonization by Proteobacteria [176,177], suggesting a possible role of UA as a modulator of nutritional status for therapeutic intervention in obesity. Further, UA decreases the ratio of Firmicutes:Bacteroidetes and enhances the growth of SCFAs-producing bacteria in hypercholesterolemic hamsters [178], while reducing plasma cholesterol (≈16%) and inhibiting intestinal cholesterol absorption (2.6–9.2%). UA intake also modifies the microbiota of *Drosophila melanogaster*, while contributing to their life span extension [179]. Moreover, several preclinical studies about the link of liver fibrosis from the UA-treatment effects to its activity in the gut microbiota [180,181,182]. In liver fibrosis, the diversity and abundance of gut microbiota are reduced, and the bacterial composition is unbalanced, reducing potentially beneficial groups as Firmicutes. UA-treatment ameliorates the microbiota dysbiosis developed by liver fibrosis in mice, increasing the gut microbiota diversity and abundance and the number of potentially beneficial bacteria and balancing the microbiota composition [183]; however, the bacterial changes in the ilea and feces are not consistent [180]. The balancing effect of UA on bacterial dysbiosis related to liver fibrosis has also been confirmed by sequencing of the 16S rRNA gene. Potential beneficial bacteria, such as those belonging to phylum Firmicutes and the genera *Lactobacillus* and *Bifidobacterium*, are increased in animals supplemented with UA [181]. UA also improves intestinal dysbiosis together with the expression of the tight junction proteins Claudin 1 and Occludin in the ileum of rats [182]. This study postulate that the protective effect of UA on the mucosal barrier of the gut in rats with liver fibrosis is related to the inhibition of intestinal NOX-mediated oxidative stress [182].

The limited evidence currently available from pre-clinical studies suggests that the effects of oleanolic acid, as well as those of ursolic acid, on gut microbiota, are related, at least in part, to its activity on the bile acid receptor TGR5 [184,185,186]. TGR5 is a G protein-coupled membrane receptor found mainly in the intestine. TGR5 is responsible for the release of the glucagon-like peptide 1 (GLP-1: a potent glucose-dependent insulin-stimulating hormone) and peptide YY (PYY: a potent appetite-regulating hormone that belongs to the neuropeptide Y and pancreatic polypeptide family of peptides), via the Gs/cAMP/PLC/Ca2+ pathway [184]. TGR5-mediated GLP-1 and PYY release is inhibited by H2S, which is generated by sulfur-reducing bacteria (SRB) from the gut microbiota [184]. The SRB are able to use H2 and organic compounds, such as lactate and formate, as electron donors for the reduction in sulfate or other oxidized sulfur compounds to generate H2S [187]. In the human colon, SRB are predominantly members of *Desulfovibrio* genus in the class δ-Proteobacteria [187,188].

### 7.7. Effects on Inflammatory Mediators

Investigations in cell cultures and experimental animals have demonstrated that treatments with pentacyclic triterpenoids significantly inhibited the pro-inflammatory cytokine production in a number of inflammation and oxidative stress scenarios. For instance, OA and UA reduce the TNF-α induced E-selectin expression in human umbilical vein endothelial cells (HUVEC) [189], inhibiting IL-6 release in lipopolysaccharide activated Mono Mac 6 cells [190] and suppress the endothelin-1 pathway in Zucker diabetic rats [102].

Pan et al. [94] showed that the application of 40 µM OA to HUVEC recovered the levels of angiotensin 1–7, NO and eNOS, downregulated by ox-LDL. The OA treatment was also able to reduce IL-1β, TNF-α and IL-6 levels. These same OA-elicited responses were obtained in New Zealand white rabbits fed a high-fat diet. Wang et al. [95] evaluated the effects of intraperitoneally administered OA (20 mg/kg/day for two weeks) on hepatic insulin resistance in Lepdb/db obese diabetic mice and observed a decrease in lipid accumulation, cellular and mitochondrial ROS production as well as in the levels of IL-1β, IL-6 and TNFα, whereas enhanced insulin signaling and inhibited gluconeogenesis. An et al. (2017) investigated the effect of OA on the carotid artery injury in STZ-induced diabetic male Sprague–Dawley rats [191]. The administration of the triterpene (100 mg/kg/day) improved glucose metabolic, neointimal hyperplasia and endothelial dysfunction. In addition, it downregulated serum levels of TNF α, IL-1β, IL-6 and IL-18. Consistent with the serum outputs, it was demonstrated that OA caused a significant inhibition of nucleotide-binding domain, leucine-rich containing family, pyrin domain containing 3 (NLRP3) inflammasomes signaling pathway, caspase 1 and IL1β in the carotid artery of diabetic rats. A reduction in plasma levels of IL-6 and TNFα by OA was likewise observed by Gamede et al. [96] in high fat-high carbohydrate diet-induced prediabetic male Sprague–Dawley rats. In agreement with these results, a significant reduction in the gene expression of IL-1β and IL-6 in the liver and adipose tissue, as well as of TNFα in adipose tissue of HFD-induced obese C57BL/6J female mice was observed by Djeziri et al. [151]. Obese mice fed a high-fructose diet presented increased RNA expression of TNFα and MCP-1 in white adipose tissue and liver, which was ameliorated after being fed pomace olive oil rich in tritepenes, including OA [96,152,161]. In addition, Gamede et al. [153] found decreased plasma proinflammatory cytokines, including TNF-α and IL-1β in Sprague Dawley rats fed a high fat, high carbohydrate diet after a 20-week treatment with OA.

Several pro-inflammatory cytokines that adversely affect insulin signaling are regulated by the transcription factor nuclear factor (NF)-kB. NF-kB is triggered by endogenous and exogenous stimuli via phosphorylation of the inhibitory subunit IkB by IKK. Several natural triterpenes inhibit IKK and block NF-kB activation. OA also irreversibly inhibits the inflammatory enzyme phospholipase A2 at µmolar concentrations [161].

## 8. A Unifying Hypothesis of the Oleanolic Acid Pharmacological Activity

Oleanolic acid is a natural component of many common plant foods (e.g., apples, pears, tomato, grapes, olives, ginger, mango, strawberries and olive oil) as well as culinary and medicinal herbs (e.g., rosemary, oregano, sage, basil, fennel, ginseng and olive leaf) [192,193]; therefore, OA is an integral part of the human diet. OA has been considered as a low-bioavailable biomolecule due to its very low solubility in water [78], although recent research has demonstrated that OA bioavailability may be highly increased when administered dissolved in lipophilic matrices [82,83]. Human studies have shown that OA bioavailability differs when administered as a pure compound or as part of a complex matrix, such as food. The presence of fat appears of major importance since OA solubilization and micellization within lipophilic compounds are necessary prior to its absorption [194]. Evidence indicates that, after passing through the liver, OA is thoroughly distributed in the tissues and circulates in the bloodstream mainly associated with plasma albumin, although levels of OA are also detected in the form of free acid [84]. Animal and human assays have demonstrated that OA is therapeutically effective without apparent side effects [195,196,197].

In the last decades, the progressive aging of the population and a sedentary lifestyle have triggered overweight/obesity and the prevalence of chronic metabolic diseases, in which oxidative stress, inflammation and alterations in the signaling pathways of insulin play a leading role. In this sense, OA is endowed with a wide range of pharmacological properties with beneficial effects against a variety of these prevalent degenerative diseases. In this review, we have highlighted that OA improves lipid metabolism and obesity in animal models of obesity and hyperlipidemia. This triterpene reduces plasma concentration of FFA, TG, TC and LDL, and slightly increases HDL [90,91,92,93,96]. In addition, declines ghrelin and rises leptin serum levels. Furthermore, OA downregulates a number of lipogenic genes, including ACC, ACAT, AdipoR1, AdipoR2, PPAR-α and PPAR-γ [90,101]. As global results, OA reduces body weight, visceral fat, liver weight [90,96] and diminishes hepatic microvesicular steatosis and lipid droplets [91]. Dyslipidemia can also trigger systemic inflammation and insulin resistance via changes in the populations of gut microbiota [168,169], probably through LPS-induced inflammatory endotoxemia [170]. Short-chain fatty acids resulting from the fermentation of dietary fiber by bacteria and bile acids might also be possible mediators linking gut bacteria to adipose tissue dysfunction and insulin resistance [171]. Despite the fact that the effects of OA on gut microbiota have been scarcely studied, existing data revealed that OA stimulates *Lactobacillus* growth in vitro [172]. Likewise, in a murine model of autoimmune myocarditis, OA recovers the fecal population of Muribaculaceae, a prevalent and abundant bacterial component of the gut microbiome of mammals [173]. These preservative activities of OA have been related to the anti-inflammatory and antioxidant effects of triterpene. In addition, the research with bardoxolone methyl in rodents shows that this synthetic derivative of OA is able to re-balance the proportions of Bacteroides/Prevotella, *Bifidobacterium* spp. and *Lactobacillus* spp. in cecal content [174], which were altered by a high-fat diet [174].

OA also improves glycemic control, and it does so on at least four levels: the modulation of complex carbohydrate metabolism through the inhibition of α-glucoside enzymes [56,108,109], the enhancement of insulin signaling cascades [122,123,124,125,126,127,128,129,130,131,132,133,134,135,136,137,138,139,140], the preservation of beta-cell functionality [110,111,112,113,114,115,116,117,118,119,120,121] and the inhibition of AGEs formation in not insulin-sensitive tissues [137,155,156]. In human and animal assays, the OA intake reduces fasting blood glucose, fasting serum insulin, HbA1c, HOMA-IR score [98,148,150], pentosidine, Nε-(carboxymethyl)lysine and urinary glycated albumin [156].

Different studies performed with hypertensive animals have demonstrated that OA reduces both systolic blood pressure (SBP) and mean arterial pressure (MAP) and increases the urinary volume and urine sodium excreted [144,146]. This antihypertensive effect has been attributed to the regulation of liver lipid metabolites at both protein and mRNA levels [99].

Obesity, dyslipidemia, hypertension and hyperglycemia are pathophysiological conditions risk factors included in metabolic syndrome, which might lead to type-2 diabetes mellitus, cardiovascular disease, non-alcoholic fatty liver disease, neurodegenerative disorders and some types of cancer [141,142,143]. Although many mechanisms have been proposed to underlie their effects, a unifying proposal is that oxidative stress and inflammation represent a common pathway of injury. In this review, we have presented evidence indicating that OA plays an efficient role in preventing lipid peroxidation and superoxide anion accumulation [159] and reinforces the endogenous intracellular antioxidant. On the other hand, investigations in cell cultures and experimental animals have demonstrated that OA significantly inhibits pro-inflammatory cytokine production (IL-1β, IL-6, IL-8, TNF-α, MCP-1) in a number of inflammation and oxidative stress scenarios [102,189,190]. Consistent with the serum outputs of inflammatory cytokines, OA causes a significant inhibition of nucleotide-binding domain, leucine-rich containing family, pyrin domain containing 3 (NLRP3) inflammasomes signaling pathways [191]. In addition, there is experimental data showing that OA recovers NO, iNOS and eNOS in inflammatory states [94,198].

OA is a versatile molecule, which directly reacts with single proteins involved in carbohydrate and lipid metabolisms, antioxidant and anti-inflammatory responses as well as insulin signaling; however, the major contributions of this triterpene against chronic diseases associated with metabolic syndrome derives from its ability to modify the expression of key genes in the adaptive cellular response against the oxidative and chemo-toxic stresses, with a leading role for the interaction with the transcription factors Nrf2 and NFκB [199]. The transactivation of Nrf2 by OA stimulates transcription of antioxidant enzymes as well as of genes involved in GSH biosynthesis and regeneration and in the production of NADPH. Likewise, the improvement of dyslipidemia by OA seems to be mediated, at least partially, by this transcription factor since both SREBP-1 and PPAR-γ are direct targets of Nrf-2 [199]. The upregulation of the glyoxalase system and the repression of the polyol pathway and AGEs generation by OA are also effects that may proceed via Nrf-2 activation since the glyoxalase gene carries a functional ARE in exon 1 with the ability to bind Nrf2 [200]. On the other hand, OA inhibits IκB kinase activity, blocking the NF-kB activation and, therefore, the inflammatory response and cell death. Thus, OA damps both oxidative stress and inflammation, acting on these two opposite pathways.

The introduction of urgent preventive measures that diminishes the obesity pandemic and delays or decreases the incidence of prevalent chronic metabolic diseases is of capital importance for national health systems worldwide. In this sense, the use of functional foods containing OA is, undoubtedly, an interesting alternative. Further investigations will be necessary to extend the use of this natural triterpene in the design of new drugs and foods, which allow personalized diets and nutrigenomic approaches for the prevention of high-prevalence chronic disorders such as neurodegenerative and cardiovascular diseases, cancer or diabetes.

## Figures and Tables

**Figure 1 nutrients-14-00623-f001:**
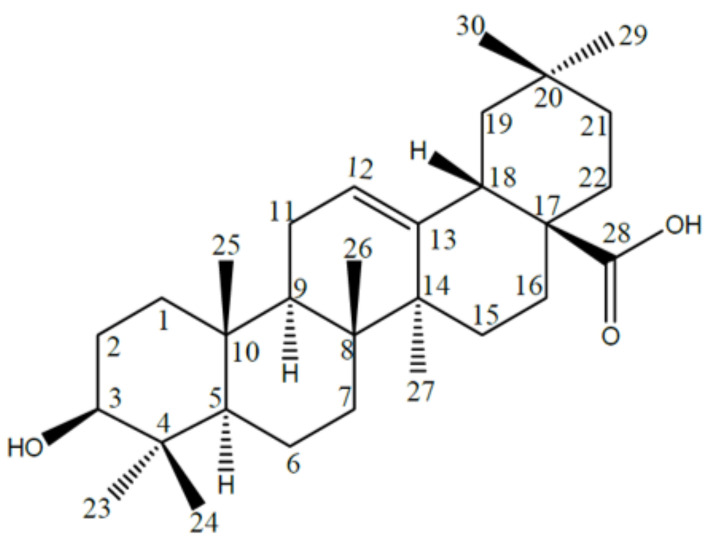
Chemical structure of oleanolic acid.

**Figure 2 nutrients-14-00623-f002:**
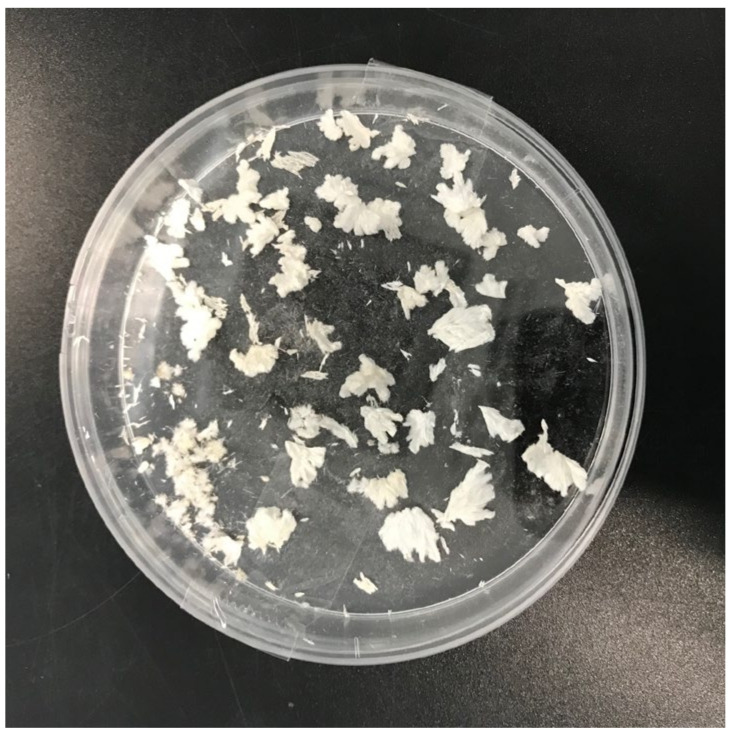
Oleanolic acid in the form of prismatic crystals.

**Figure 3 nutrients-14-00623-f003:**
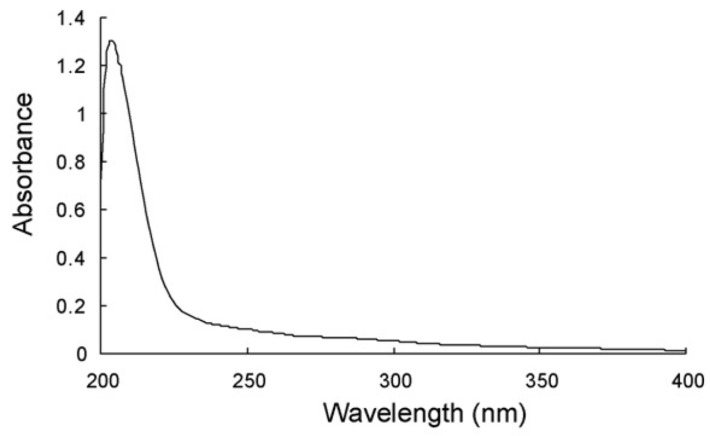
Absorbance spectrum of oleanolic acid.

**Figure 4 nutrients-14-00623-f004:**
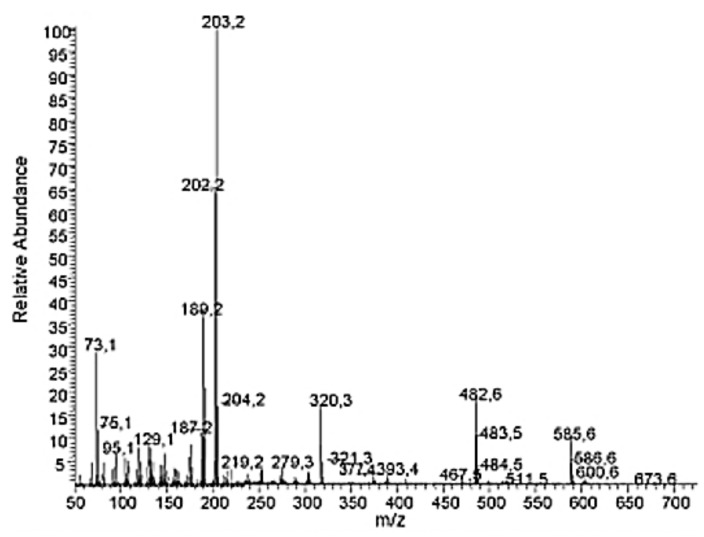
Mass spectrum of oleanolic acid, showing five major m/z fragments: 203.2; 202.2; 189.2; 73.1; 187.2.

**Table 1 nutrients-14-00623-t001:** Pharmacokinetic parameters of oleanolic acid (OA) in human serum after oral single-dose administration.

OA Dose (mg)	OA Formulation	Participants	n	C_max_ (ng/mL)	t_max_ (h)	t_1/2_ (h)	AUC_0→t_ (ng h/mL)	CL (L/h)	V_d_ (L)	Ref.
40	capsules	healthy males	18	12.1 ± 6.8	5.2 ± 2.9	8.7 ± 6.1	114.3 ± 74.9	555.3 ± 347.7	3371.1 ± 1990.1	[80]
20	normal tablets	healthy males	18	18.9 ± 8.0	2.9 ± 1.2	12.3 ± 9.0	112.2 ± 56.6	No data available	No data available	[81]
20	dispersable tablets	healthy males	18	17.8 ± 7.5	2.5 ± 1.0	16.1 ± 12.1	109.7 ± 41.6	No data available	No data available	[81]
30	OA-enriched ^a^POO	healthy males	9	598.2 ± 176.7	3.0 ± 0.8	4.6 ± 0.1	3025.2 ± 847.1	35.1 ± 4.2	81.4 ± 9.7	[82]
4.7	^b^FOO	healthy adults (6 m, 6 f)	12	5.1 ± 2.1	4 (1–6) ^c^	4.1 ± 0.2	28.2 ± 9.4	No data available	No data available	[83]

Data expressed as mean ± standard deviation except for t_max_, which is expressed as median (min-max). C_max_, serum maximal concentration; t_max_, time to maximal concentration; t_½_, elimination half-life; AUC_0→t_, area under the curve from 0 to the last measurement; CL, clearance; V_d_, volume of distribution. ^a^POO, pomace olive oil; ^b^FOO, high phenolics-high triterpenes functional olive oil. ^c^ All data in the table are expressed as mean ± standard deviation except for t_max_, in [83] which is expressed as median (min-max range).

## Data Availability

Not applicable.

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
