# Peer review of "Oleanolic Acid: Extraction, Characterization and Biological Activity"

_nutrients, 2022, doi:10.3390/nu14030623_

Round 1

Reviewer 1 Report

We know that OA is very important for food and health, and this article is a very nice compilation of data that gives an almost complete picture of current knowledge on the composition, extraction and bioactivity of OA. This work will no doubt be very useful for food and pharmaceutical developers.

Yet I have some suggestions to give greater value to this work.

Lines 68-69 : The only original input from the authors would be Figure 2 representing prismatic crystals of OA. I suggest adding (or replacing) a figure with a larger zoom of the crystals and also, if possible, a parallel figure showing the amorphous nature obtained by the fast cooling.

Lines 408-437 : several publications are cited and discussed to give plasma OA concentrations based on different doses of administration. Would it be possible to synthesize all this data in an original graph and/or table?

Lines 456-520 : OA effects on lipid profile is a section that can be divided (like the following sections) into several subsections (for example: effect on gene expression, effect on visceral fat and hormones, body weight and TG level, animal models, human effects…)

Lines 521-641 : I suggest that all sections presented in the different paragraphs here could be included in a single section (for example OA effects on carbohydrate metabolism/ or carbohydrate profiles, to be symmetric with the precedent one). This while keeping the initial cutting: alpha-glucosidases, pancreatic functions, insulin gene transcription, B-cells survival, insulin signaling with the different activations and inhibitions as already drafted.

Lines 749-817: to verify that all Latin names are in italic

Line 856: I suggest that a summary of the effects of the OA should be included here, with the most important information from each section on bioactivities. A figure (graph or drawing) and a hypothesis of their action would also be the most welcome. Similarly, a validation or own experimentation if the authors have it.

Author Response

Thank you for your kind letter. We greatly appreciate your comments as well as those of the reviewers. The manuscript has been revised in consonance with the comments and recommendations. We are grateful to the reviewers for the thorough way they have assessed our manuscript. We have answered to all their queries, and we believe that the modifications have significantly improved the paper.

Reviewer 1:

We know that OA is very important for food and health, and this article is a very nice compilation of data that gives an almost complete picture of current knowledge on the composition, extraction and bioactivity of OA. This work will no doubt be very useful for food and pharmaceutical developers.

Yet I have some suggestions to give greater value to this work.

Lines 68-69 : The only original input from the authors would be Figure 2 representing prismatic crystals of OA. I suggest adding (or replacing) a figure with a larger zoom of the crystals and also, if possible, a parallel figure showing the amorphous nature obtained by the fast cooling.

Response: We agree with the reviewer that a larger image would be helpful. Unfortunately, this is the only image we have. It is actually quite old and at that time resolution was not so good.

Lines 408-437 : several publications are cited and discussed to give plasma OA concentrations based on different doses of administration. Would it be possible to synthesize all this data in an original graph and/or table?

Response: We have added Table 1 to the manuscript following the reviewer’s recommendation.

Lines 456-520 : OA effects on lipid profile is a section that can be divided (like the following sections) into several subsections (for example: effect on gene expression, effect on visceral fat and hormones, body weight and TG level, animal models, human effects…)

Response: We have renumbered this section following the reviewer’s recommendation.

Lines 521-641 : I suggest that all sections presented in the different paragraphs here could be included in a single section (for example OA effects on carbohydrate metabolism/ or carbohydrate profiles, to be symmetric with the precedent one). This while keeping the initial cutting: alpha-glucosidases, pancreatic functions, insulin gene transcription, B-cells survival, insulin signaling with the different activations and inhibitions as already drafted.

Response: We have presented these subsections in a single one and we have renumbered the section following the reviewer’s recommendation.

Lines 749-817: to verify that all Latin names are in italic

Response: All Latin names have been checked and written in italics those indicated by the taxonomic rules.

Line 856: I suggest that a summary of the effects of the OA should be included here, with the most important information from each section on bioactivities. A figure (graph or drawing) and a hypothesis of their action would also be the most welcome. Similarly, a validation or own experimentation if the authors have it.

Response: A whole new section (8) has been added to the end of the manuscript summarizing the content of the review.

Reviewer 2 Report

This review written well and produced salient points of research updated systematically and compromised to bioactive compound specific and nutraceutical driven illustration. The authors focus on the impact of Oleanolic Acid (OA) on physico-chemical properties, value of phytochemicals and valuable food source such as plant metabolites and overview of extraction methods and integration of pharmacological aspects in particular metabolic diseases including hyperlipidemia and diabetes, and obesity. Include interaction with microbiome changes. It is a valuable report that represents Nutrients and Journal Aims to the public that could highlight that the essential element of a nutrient chemical might be visualized with its biological aspects from nature entity and yield profile depending on extraction methods and characterization with structural and functional instrumentation. Hence, I agree that it make a substantial contrition to inspire research community go through J of Nutrients.

Minor typos need to be corrected

Ln 200 55 OC >> 55OC

Ln 306, it need to confirm it, 28,2 MPa (megapascals)

Ln 474 Lepdb/db obese >> Lepr Db/db obese

Ln 489 20 mg/kg bw >> BW

Ln 599 Type [124]2 >> Type 2 [124]?

Ln 613 FoxO1 >> FOXO1

Ln 695 Co adyvant>> adjuvant

Ln 739 tert-butyl hydroperoxide

Ln 769 make consistent symbol, IL-1a and IL-1b or IL-1α and IL-1β

Author Response

Thank you for your kind letter. We greatly appreciate your comments as well as those of the reviewers. The manuscript has been revised in consonance with the comments and recommendations. We are grateful to the reviewers for the thorough way they have assessed our manuscript. We have answered to all their queries, and we believe that the modifications have significantly improved the paper.

Ln 200 55 OC >> 55OC

Response: This typo has been corrected

Ln 306, it need to confirm it, 28,2 MPa (megapascals)

Response: The value is actually correct but the comma has been substituted with a dot. The correct value is 28.2 Mpa.

Ln 474 Lepdb/db obese >> Lepr Db/db obese

Response: This typo has been corrected

Ln 489 20 mg/kg bw >> BW

Response: This typo has been corrected

Ln 599 Type [124]2 >> Type 2 [124]?

Response: This typo has been corrected

Ln 613 FoxO1 >> FOXO1

Response: This typo has been corrected

Ln 695 Co adyvant>> adjuvant

Response: This typo has been corrected

Ln 739 tert-butyl hydroperoxide

Response: This typo has been corrected

Ln 769 make consistent symbol, IL-1a and IL-1b or IL-1α and IL-1β

Response: Al symbols have been checked and corrected.